# MoS_2_/SnS/CoS Heterostructures on Graphene: Lattice-Confinement Synthesis and Boosted Sodium Storage

**DOI:** 10.3390/molecules28165972

**Published:** 2023-08-09

**Authors:** Ruyao Zhang, Yan Dong, Yu Su, Wenkai Zhai, Sailong Xu

**Affiliations:** 1State Key Laboratory of Chemical Resource Engineering, Beijing University of Chemical Technology, Beijing 100029, China; 2022210730@buct.edu.cn (R.Z.); dongyan2232021@163.com (Y.D.); suyu_yh@163.com (Y.S.); 2021210568@buct.edu.cn (W.Z.); 2Quzhou Institute for Innovation in Resource Chemical Engineering, Quzhou 324000, China

**Keywords:** layered double hydroxide precursor, crystalline lattice confinement, multiple sulfide heterostructures, anode nanomaterials, sodium-ion batteries

## Abstract

The development of high-efficiency multi-component composite anode nanomaterials for sodium-ion batteries (SIBs) is critical for advancing the further practical application. Numerous multi-component nanomaterials are constructed typically via confinement strategies of surface templating or three-dimensional encapsulation. Herein, a composite of heterostructural multiple sulfides (MoS_2_/SnS/CoS) well-dispersed on graphene is prepared as an anode nanomaterial for SIBs, via a distinctive lattice confinement effect of a ternary CoMoSn-layered double-hydroxide (CoMoSn-LDH) precursor. Electrochemical testing demonstrates that the composite delivers a high-reversible capacity (627.6 mA h g^−1^ after 100 cycles at 0.1 A g^−1^) and high rate capacity of 304.9 mA h g^−1^ after 1000 cycles at 5.0 A g^−1^, outperforming those of the counterparts of single-, bi- and mixed sulfides. Furthermore, the enhancement is elucidated experimentally by the dominant capacitive contribution and low charge-transfer resistance. The precursor-based lattice confinement strategy could be effective for constructing uniform composites as anode nanomaterials for electrochemical energy storage.

## 1. Introduction

Rechargeable sodium-ion batteries (SIBs) were increasingly promising in the field of large-scale electric energy storage for renewable energy and smart grids during the “beyond-lithium-ion-batteries” era, due to the advantageous characteristics of low cost, high abundance, and wide distribution of sodium resources [1,2,3,4]. The larger radius of Na^+^ (1.02 Å) than that of Li^+^ (0.76 Å), however, results in the sluggish kinetics of insertion/extraction of Na^+^ into active materials [5,6,7] and thus hinders the development and application of SIBs. The limitation has triggered a great deal of effort to develop numerous advanced anode nanomaterials for SIBs, such as carbon-based nanomaterials [8,9,10,11], metallic alloys [12,13,14], two-dimensional (2D) metal carbides (MXenes) [15,16,17,18], metal dichalcogenides [19,20,21,22], and metal sulfides. In particular, metal sulfides are considered to be favorable for sodium storage reactions due to their weak metal–sulfur (M–S) bonds, good reversibility and high capacity [23]. Cobalt sulfide (CoS) is an interesting semiconductor material and has broad applications in supercapacitors, lithium-ion batteries, and catalysis [24,25,26], as well as SIBs due to its high-theoretical capacity [27]. However, CoS still suffers from poor cycling performance due to large volume-change-induced electrode pulverization, and also from poor rate performance owing to the sluggish electrode kinetics [28].

To alleviate the problems, effective approaches to improve electrochemical performance mainly focus on hybridizing with carbonaceous materials [29,30,31], designing nano- and sub-nano scale structures [32,33], and/or integrating with other metal sulfides [27,34,35]. Using glucose-assisted hydrothermal reaction and subsequent calcination treatment [36], carbon coated MoS_2_/CoS nanostructures on carbon cloth (MoS_2_/CoS@CC) were fabricated, and achieved a reversible capacity of 605 mA h g^−1^ after 100 cycles at 0.5 A g^−1^ and rate performance of 366 mA h g^−1^ at 8.0 A g^−1^. Using a sol–gel method [37], CoS/Co_9_S_8_ quantum dots-embedded N/S-doped carbon nanosheets were prepared, and afforded rate capability (600 and 500 mAh g^−1^ at 0.2 A g^−1^ and 10 A g^−1^, respectively) and a capacity retention of 87% after 200 cycles at 1 A g^−1^. From cobalt-containing metal–organic frameworks (MOFs) of ZIF-67 precursor [38], CoS/Co_9_S_8_ heterostructures embedded within N-doped carbon were synthesized, and provided a capacity of 409.2 mAh g^−1^ after 100 cycles at 0.1 A g^−1^ and rate capability of 134.9 mAh g^−1^ at 10 A g^−1^. By combining hydrothermal synthesis and annealing treatment [39], binary metal sulfide (CoS@SnS) heterostructure confined within carbon microspheres were acquired, and delivered reversible capacity of 463 mAh g^−1^ and long durability of 368 mAh g^−1^ after 1000 cycles at 2 A g^−1^. Using a solvothermal synthesis and polydopamine (PDA) encapsulation [40], binary CoS/Cu_2_S nanocages within nitrogen-doped carbon shell (CoS/Cu_2_S@N-C) were constructed, and displayed a capacity of 435.3 mAh g^−1^ after 1000 cycles at 2 A g^−1^ and high-rate capacity of 347.2 mAh g^−1^ after 2300 cycles at 10 A g^−1^. All the above anode nanomaterials exhibited highly enhanced electrochemical performances. The enhancements were contributed from the well-designed nanostructures and their homogeneous distributions, as well as hybridization with a high electron conductive carbon/multiple sulfide heterostructure [41], which were achieved mainly via confinement strategies of surface templating and/or three-dimensional encapsulation.

Herein, we report a multi-component MoS_2_/SnS/CoS heterostructures/graphene (MoS_2_/SnS/CoS/rGO) composite as a high-rate anode nanomaterial for SIBs. The well-dispersed sulfide composite is derived by pyrolyzing a single-source ternary CoMoSn-layered double-hydroxide (CoMoSn-LDH) precursor via a distinctive lattice confinement effect (Figure 1). LDHs are well-known as one large type of two-dimensional Mg(OH)_2_-like layered structures that consist of type-and-ratio-tunable metallic cations and anions [42]. The metallic cations are well-orderedly dispersed within the same crystalline layers [42], that is, the synthesis process and crystal growth of one metal cation are confined by, and well-distributed within, the matrix of the other metal cations, which provides a lattice confinement-based strategy to design and prepare well-distributed nanocomposites, such as alloy [43] and mixed metal oxides [44] in others’ studies, as well as sulfides in our recent studies [45,46,47]. As a result, the MoS_2_/SnS/CoS/rGO composite delivers a greatly boosted reversible specific capacity at 0.1 A g^−1^ and long-cycled rate capacity at 5.0 A g^−1^, exceeding those of the single-, bi- and mixed sulfides. The enhancement is further elucidated experimentally by the interfacial interaction between heterostructures, dominant capacitive contribution, and low charge-transfer resistance.

## 2. Results

The CoMoSn-LDH precursor/GO was prepared using a traditional co-precipitation method. The resulting X-ray diffraction (XRD) pattern of the precursor shows the characteristic basal (00*1*) diffraction peaks and non-basal (012) diffraction peak (Appendix A), reflecting the formation of ternary CoMoSn-LDH precursor. The scanning electron microscope(SEM) image further confirms that the precursor was grown on the supporting GO (Appendix A). Subsequently, the CoMoSn-LDH precursor/GO was subjected to the calcination and sulfurization at 300 °C for 2 h in Ar. The resulting product (Figure 2a) shows the diffraction peaks at 14.4° and 32.7°, which are assigned to the (002) crystal plane of hexagonal 2H-MoS_2_ (PDF #37-1492); and the diffraction peaks at 22.0°, 26.0°, 27.5°, 31.5°, 39.0°, 44.7° and 56.7°, which are attributed to the crystal planes of (110), (120), (021), (111), (131), (141) and (042) for SnS (PDF #39-0354), respectively. The other diffraction peaks at 30.6°, 35.3°, 46.9° and 54.4° correspond to the (100), (101), (102) and (110) crystal planes of CoS (PDF#39-0354), respectively. All the aforementioned XRD results demonstrate the co-existence of the MoS_2_, SnS, and CoS phases in the composite. Moreover, the SEM image (Appendix A) shows that numerous nanosheets are dispersed on the rGO support, and the energy dispersive spectrometer (EDS) mapping images for SEM shows the uniform distribution of the elements of Co, Mo, Sn, S and C in the MoS_2_/SnS/CoS@rGO composite (Appendix A).

Furthermore, the high-resolution transmission electron microscopy (TEM) reveals the MoS_2_ nanosheets are confined on rGO support (as shown by a checkmark in Figure 2b). Distinct interfaces are also identified between MoS_2_, SnS, and CoS (as shown by two dotted boxes in Figure 2c), which strongly suggests the lattice confinement of the CoMoSn-LDH precursor. Three different lattice fringes are quantitatively determined: a lattice spacing of 0.62 nm corresponding to the (002) plane of MoS_2_ (Figure 2c), 0.19 nm to the (102) plane of CoS, and 0.28 nm to the (111) plane of SnS (Figure 2d).

The chemical composition and element valence of the composite were characterized by X-ray photoelectron spectra (XPS). Figure 3a shows a XPS full spectrum that demonstrates the presence of Co, Mo, Sn, C, S, and O elements. The high-resolution Co 2p spectrum was divided into two spin-orbit doublets and two oscillatory satellite peaks (Figure 3b). The former spin-orbit doublets with binding energies at 779.2/780.7 eV and 794.2/795.8 eV correspond to Co 2p_3/2_ and Co 2p_1/2_, respectively, which are assigned to the Co^2+^/Co^3+^ redox pair; while the latter satellite peaks at 786.5 and 802.1 eV are related to Co 2p_3/2_ and Co 2p_1/2_, respectively. The Sn 3d spectrum shows two strong peaks at 487.4 eV and 495.8 eV, which are assigned to Sn 3d_5/2_ and Sn 3d_3/2_, respectively, and thus proves that SnS mainly exists in a state of Sn^2+^ (Figure 3c). The Mo 3d spectrum was fitted by two peaks at 229.3 eV and 232.4 eV, which are assigned to Mo^4+^ 3d_5/2_ and Mo^4+^ 3d_3/2_, respectively, and also demonstrates that Mo exists in a tetravalent form. Another peak at 235.5 eV was also visible, indicating that the partial Mo^4+^ at the surface was oxidized to Mo^6+^ due to the exposure to air (Figure 3d). The S 2p spectrum was de-convoluted into two peaks at 162.2 eV and 163.3 eV (Figure 3e), which are attributed to S 2p_3/2_ and S 2p_1/2_, respectively. Note that one peak at 164.5 eV was observed, which indicates the existence of the C–S bond in the composite; this result indicates that the rGO carbon substrate is tightly bound to the three metal sulfide materials, which is more favorable for electronic migrate and maintaining structural stability. Moreover, the high-resolution C 1s spectrum was fitted by those four peaks at 284.8 eV (corresponding to C–C), 285.8 eV (C–S), 287.2 eV (C–O), and 289.5 eV (O–C=O) (as shown in Figure 3f).

The carbonaceous graphitization degree was estimated using the Raman spectroscopy (Appendix A). Two strong peaks are visible at 221.0 cm^−1^ and 310.8 cm^−1^, which reflect the presence of SnS and MoS_2_, respectively. In particular, two pronounced peaks appear at 1340.6 cm^−1^ and 1596.7 cm^−1^, which are attributed to the carbonaceous D band (carbon atom lattice defect) and G band (carbon atom sp^2^ hybridization), respectively. The value of intensity ratio (I_D_/I_G_) was thus calculated to be 1.21, which is indicative of a relatively low degree of graphitization. Furthermore, the exact carbon content in the composite was examined using Thermal gravimetric (TG) analysis. Two main stages of weight loss were clearly visible (Appendix A): the first stage occurred before 450 °C, which involves both the removal of water and the oxidation of metal sulfide MoS_2_/SnS/CoS; the second stage ranged between 400 °C and 670 °C, which corresponds to the combustion of carbon. As a result, the carbonaceous content in the MoS_2_/SnS/CoS@rGO composite was calculated to be 18.6 wt%. In addition, the specific surface area and pore size distribution of MoS_2_/SnS/CoS@rGO were quantitatively determined by measuring N_2_ adsorption and desorption isotherms. The specific surface area of the composite was calculated to be 30.3 m^2^ g^−1^ according to the Brunauer–Emmett–Teller (BET) method (Appendix A). The pore size distributions ranged from 3 and 10 nm (Appendix A), clearly reflecting the mesoporous feature. The mesoporous size distribution, together with the relatively low specific surface area (30.3 m^2^ g^−1^), is well-recognized to hold the electrolyte and facilitate the Na^+^ diffusion, and thus to improve rate performance [48,49]. As a result, by combining the above-obtained characterization results, the MoS_2_/SnS/CoS@rGO composite was successfully obtained, and had the advantageous characteristics capable of boosting the sodium-storage performance: well-dispersed multi-component MoS_2_/SnS/CoS with the interfaces, highly electron conductive rGO support (18.6 wt%), appropriate specific surface area (30.3 m^2^ g^−1^) and mesoporous size distribution.

The electrochemical performance of MoS_2_/SnS/CoS@rGO was evaluated by measuring cyclic voltammetry (CV) curves (Appendix A) at a scan rate of 0.1 mV s^−1^ between 0.01 and 3.0 V. During the first reduction scan, the first reduction peak at 0.78 V is attributed to both the decomposition of the electrolyte and the formation of irreversible solid-electrolyte interface (SEI). Three peaks were observed at 0.21 V, 0.58 V, and 0.91 V, corresponding to the alloying reaction of Sn and Na and indicating that the alloying reaction is a multi-step process. Another reduction peak at 1.19 V is attributed to the process of converting from MoS_2_ to Na_x_MoS_2_. During the oxidation sweep, two peaks at 0.18 V and 0.70 V were visible, which are assigned to the multi-step Na-Sn dealloying reaction. The subsequent two broad peaks at 1.00 V and 1.37 V appeared, due to the conversion reaction of Sn and Na_2_S to SnS. Another oxidation peak at 1.82 V appeared, owing to the reversible transformation reaction between metallic Co, Mo and Na_2_S matrix. Consequently, the mechanism reactions for sodium storage are presented in the following equations with Gibbs (ΔG, kJ/mol).
Na_x_MoS_2_ + (4 − x) Na^+^ + (4 − x) e^−^ ⇌ Mo + 2Na_2_S  ∙∙ (ΔG = 58.672)(1)
Na_x_MoS_2_ + (4 − x) Na^+^ + (4 − x) e^−^ ⇌ Mo + 2Na_2_S  ∙∙ (ΔG = 58.672)(2)
CoS + 2Na^+^ + 2e^−^ ⇌ Co + Na_2_S  ∙∙∙∙∙∙∙∙∙∙∙∙∙∙∙∙∙∙∙∙∙∙∙∙∙∙∙∙∙ (ΔG = 53.461)(3)
4Sn + 15Na^+^ + 15e^−^ ⇌ Na_15_Sn_4_  ∙∙∙∙∙∙∙∙∙∙∙∙∙∙∙∙∙∙∙ (Alloying reaction)(4)

The first three galvanostatic charge/discharge curves were recorded for the MoS_2_/SnS/CoS@rGO composite electrode at 0.1 A g^−1^. Figure 4a shows that the MoS_2_/SnS/CoS@rGO composite achieves the first discharge/charge capacities of 875.0 mA h g^−1^ and 742.3 mA h g^−1^, generating an initial Coulombic efficiency (ICE) of 84.8%. Figure 4b shows the comparison of cycling performance between MoS_2_/SnS/CoS@rGO, CoS@rGO, MoS_2_/CoS@rGO and MoS_2_/SnS/CoS@rGO-M, at 0.1 A g^−1^. The MoS_2_/SnS/CoS@rGO composite delivers a high-reversible capacity of 627.6 mA h g^−1^ after 100 cycles at 0.1 A g^−1^, which is superior to those of CoS@rGO (325.2 mA h g^−1^) and MoS_2_/CoS@rGO (363.4 mAh g^−1^). By contrast, the specific capacity of the MoS_2_/SnS/CoS@rGO-M rapidly decayed to less than 600 mA h g^−1^ after 80 cycles at 0.1 A g^−1^. Note that an overlap occurs between the second and third discharge/charge profiles. Further comparison of reversible capacity shows that the capacity retentions of the 5th, 10th, 11th, 100th cycles were determined to be 93.0%, 89.2%, 89.2% and 84.3%, respectively, reflecting a decrease-then-quasi-stable cycling behavior after the completion of 100 cycles at 0.1 A g^−1^ (as shown in Figure 4b). The trend is attributed to the following possibilities. (i) The formation of the SEI film and the irreversible Na^+^ intercalation/extraction, (ii) the initial pulverization and the subsequently stable MoS_2_/SnS/CoS heterostructures on the underlying rGO support. Such a type of decrease-then-stable trend was also reported early for single-atom cobalt-doped MoS_2_/carbon [50], free-standing mesoporous Co(OH)_x_/Co_2_P [51], and hollow double-shell MnS@C nanospheres [52]. The comparison strongly suggests the satisfactory cycling stability of the MoS_2_/SnS/CoS@rGO electrode, which is predominantly attributed to the synergism between the multiple sulfides.

Rate capability was further evaluated for the electrodes at densities from 0.1 to 5.0 A g^−1^. Figure 4c shows that the MoS_2_/SnS/CoS@rGO composites afforded specific capacities of 632.2, 579.1, 537.4, 513.2, 496.3, 439.8 and 371.0 mA h g^−1^ at 0.1, 0.2, 0.5, 0.8, 1.0, 2.0 and 5.0 A g^−1^, respectively; and was almost capable of returning to the initial state at 0.1 A g^−1^. However, the three electrodes of CoS@rGO, MoS_2_/CoS@rGO, and MoS_2_/SnS/CoS@rGO -M all exhibited much lower specific capacities than those of the MoS_2_/SnS/CoS@rGO composite at the corresponding current densities. In particular, at 5.0 A g^−1^, the MoS_2_/SnS/CoS@rGO composite afforded a high-reversible-specific capacity of 304.9 mA h g^−1^ after 1000 cycles (Figure 4d), with Coulombic efficiencies of ca. 100% (as shown by the golden line in Figure 4d). However, the three electrodes of CoS@rGO, MoS_2_/CoS@rGO, and MoS_2_/SnS/CoS@rGO-M exhibited rapid degradation to 150 mA h g^−1^ after 400 cycles, 180.1 mA h g^−1^ after 500 cycles, and 137.2 mA h g^−1^ after 600 cycles, respectively. As a result, the MoS_2_/SnS/CoS@rGO possesses the greatly boosted specific capacity and long-cycle stability compared with those of the CoS@rGO, MoS_2_/CoS@rGO, and MoS_2_/SnS/CoS@rGO-M. Moreover, further comparison of cycling performance is summarized in Appendix A [27,36,37,38,39,40,53,54,55,56,57], which reflects that the MoS_2_/SnS/CoS@rGO composite exhibits a comparable cycling stability to those of the sulfide anode nanomaterials reported previously. In particular, the rate performance was compared between MoS_2_/SnS/CoS@rGO and those reported sulfide anode nanomaterials for SIBS [38,39,40,55,56,57,58,59], demonstrating the excellent rate capability of the MoS_2_/SnS/CoS@rGO (Appendix A).

The above-absolved electrochemical results demonstrate that the LDH-derived MoS_2_/SnS/CoS@rGO composite exhibits greatly enhanced electrochemical performances for SIBs compared with those of the three electrodes of the individual CoS@rGO, binary MoS_2_/CoS@rGO, and the mixed MoS_2_/SnS/CoS@rGO-M. The higher enhancement is mainly attributed to the following possibilities. The multiple-component MoS_2_/SnS/CoS presents the heterostructural nanodomains and thereby plays a synergistic role in boosting the electrochemical performances. It is well-known that the interface between different components could effectively reduce the kinetic barrier of Na^+^ diffusion, and thereby provide active sites for achieving efficient secondary ion storage [60,61,62]. To support the aspect, the Nyquist plots were measured by recording electrochemical impedance spectroscopy (EIS) for the electrodes. Obviously, the MoS_2_/SnS/CoS@rGO composite exhibits a much smaller semicircle and a larger slope than those of the other electrodes (Appendix A), strongly suggesting that the former electrode has a lower charge-transfer resistance and lower Na^+^ diffusion resistance compared with those of the latter electrodes.

Furthermore, comparison of XPS spectra was conducted between the LDH-derived MoS_2_/SnS/CoS@rGO and the MoS_2_/SnS/CoS@rGO-M. The main Sn 3d of MoS_2_/SnS/CoS@rGO shift to a lower binding energy compared with those of MoS_2_/SnS/CoS@rGO-M, reflecting that there is more charge density on the Sn atoms; whereas the main Mo 3d peaks shift to a higher bonding energy compared with those of MoS_2_/SnS/CoS@rGO-M, indicating that there is less charge density on the Mo atoms. The electron transfers between SnS and MoS_2_, resulting from the interface between MoS_2_ and SnS, are due to the lattice-confinement of the LDH precursor, indeed facilitating the electrochemical performance during the charging/discharging processes.

Moreover, the Na^+^ storage kinetics of MoS_2_/SnS/CoS@rGO was examined by measuring CV curves at different scan rates from 0.2 to 1.2 mV s^−1^. Figure 5a shows that both the cathodic and anodic peaks appeared to shift to increasingly low and high potentials, respectively. As a result, the slopes of the linear relationship between the logarithmic currents (i) and scan rates (v) at the redox peak potentials were calculated to be 0.87, 0.77, 0.82, and 0.76, respectively (Figure 5b), all of which are closely approximate to 1.0 and thus indicating that the whole electrochemical process of MoS_2_/SnS/CoS@rGO is mainly controlled by capacitive contribution. Furthermore, the capacitive contribution was quantitatively determined in accordance with the formula: i = k_1_v + k_2_v^1/2^, in which where k_1_v means the fraction of capacitive contribution. Consequently, the capacitive contribution at 1.2 mV s^−1^ was 76.3% (Figure 5c). At the varied scan rates from 0.2, 0.4, 0.6, 0.8, 1.0, to 1.2 mV s^−1^, the capacitive contribution was increased from 51.0%, 56.3%, 62.3%, 67.7%, 72.2%, to 76.3% (Figure 5d). The results demonstrate that the capacitive contributions contribute dominantly to the sodium-ion storage especially at high-current densities.

## 3. Materials and Methods

### 3.1. Preparation of CoMoSn-LDH/GO Precursor

The CoMoSn-LDH/GO precursor was synthesized using a traditional co-precipitation method (as shown in Appendix A). A brown graphene sol (GO), with a concentration 6.73 mg mL^−1^, was synthesized using the modified Hummers method [63]. A 50 mg of GO sol was well dispersed into 100 mL of deionized water, and then ultrasonicated for 2 h to acquire a uniform 0.5 mg mL^−1^ sol. A mixed salt solution was prepared by dissolving Na_2_MoO_4_∙2H_2_O (2 mmol), SnCl_2_ (3 mmol) and Co(NO_3_)_2_·6H_2_O (6 mmol) into 50 mL of deionized water, respectively, followed by 2 h of ultrasonic to obtain a clear solution. Note that concentrated HCl was added drop wisely to prevent MoO_4_^2−^ from oxidizing Sn^2+^ to Sn^4+^ under the acidic condition. Subsequently, a mixed alkali solution was obtained by dissolving NaOH (40 mmol) and Na_2_CO_3_ (12 mmol) into 100 mL of deionized water under ultrasonication. The GO suspension was poured into a 500 mL four-necked flask, followed by dropping the mixed salt solution under mechanical stirring. After the completion of adding the mixed salt solution, the mixed alkali solution was also added drop wisely into the four-necked flask until the pH = 10.0 under subsequent stirring for 30 min. The as-prepared slurry was transferred quickly into a 100 mL Teflon-sealed autoclave, and kept for 24 h of aging at 120 °C. After the temperature was lowered naturally to room temperature, the CoMoSn-LDH/GO precursor was acquired by centrifugation and rinsing with the alternative deionized water/ethanol until pH = 7.0, followed by 12 h of drying in a vacuum oven at 60 °C.

### 3.2. Preparation of MoS_2_/SnS/CoS@rGO

As shown in Appendix A, both 0.2 g of CoMoSn-LDH/GO precursor and 0.2 g thioacetamide (TAA, CH_3_CSNH_2_) were dispersed into 60 mL of deionized water, followed by 30 min of magnetic stirring. The mixed dispersion was then poured into a 100 mL autoclave and maintained at 200 ° C for 24 h. After the temperature cooled to room temperature, the precipitate was collected by centrifugation and rinsed alternatively with the deionized water and ethanol, and kept for 12 h of drying at 60 °C. The MoS_2_/SnS/CoS@rGO product was obtained by subjecting the above-prepared precipitate to 120 min of calcination at 300 °C in a corundum porcelain boat in Ar with a heating rate of 5 °C min^−1^.

For comparison, MoS_2_/CoS@rGO and CoS@rGO were prepared as counterparts under the same experimental conditions without adding SnCl_2_, or Na_2_MoO_4_∙2H_2_O/SnCl_2_, respectively. A MoS_2_/SnS/CoS@rGO-M composite was directly prepared under the same experimental conditions of Na_2_MoO_4_, Na_2_[Sn(OH)_6_], Co(NO_3_)_2_∙6H_2_O and CH_3_CSNH_2_, without involving the formation of the LDH precursor.

### 3.3. Materials Characterization

The crystal structure of the sample was tested using XRD (Bruker Discover, Bruker, Berlin, Germany), with Cu Kα as the emission source (λ = 0.15418 nm). The test conditions are 10–80° and the scanning speed is 10° min^−1^. The SEM (Zeiss Supreme 55, Zeiss, Oberkochen, Germany) was used to observe the microscopic topography and size of the sample. The microstructure of the sample as well as lattice fringes were analyzed using HRTEM (JEM-2100F, JEOL, Peabody, MA, USA). The elements on the sample surface were analyzed using the XPS (Axis Supra, Shimadzu, Kyoto, Japan) radiation source as the Al target with a vacuum degree of 2 × 10^−9^ Pa; Nitrogen isothermal adsorption and desorption test (BET) uses a fully automatic adsorption instrument (ASAP-2020, Mack, Greensboro, NC, USA) to analyze the specific surface area and pore size distribution of the sample. Carbon material defects in samples were analyzed using Raman spectroscopy (RM 20001, Renishaw, West Dundee, IL, USA). The excitation wavelength is 532 nm, and the scanning range is 1000–2000 cm^−1^.

### 3.4. Electrochemical Testing

All the half cells were subjected to electrochemical testing using the CR2032 coin cells that were assembled in an Ar-filled glove box (with H_2_O, O_2_ both below 0.1 ppm). The working electrodes were acquired by mixing the active materials (MoS_2_/SnS/CoS@rGO and its counterparts), Super P and poly(vinyl difluoride) (PVDF) at a mass ratio of 7:2:1, followed by casting homogenously onto copper foils. The loading of the active material was ca. 1.5 mg cm^−2^ on each electrode foil. The counter electrode and the separator involved were sodium foil and Whatman glass fiber, respectively. The electrolyte used was a mixture consisting of NaPF_6_ (1.0 M) and ethylene glycol/dimethyl ether (DG/DME), the latter of which were prepared at a 1:1 mass ratio. Both cyclic voltammogram (CV) curves and electrochemical impedance spectra (EIS) were acquired on a commercial 760E electrochemistry workstation (Shanghai, China). The CV curves were recorded at a scan rate of 0.1 mV s^−1^ within the voltage range from 0.01 to 3.0 V (vs. Na^+^/Na), while the EIS data were obtained within the frequency range between 100 kHz and 0.1 Hz at an open circuit potential. All galvanostatic discharge/charge profiles were recorded on a LANDCT-2001A battery tester.

## 4. Conclusions

We have demonstrated the MoS_2_/SnS/CoS@rGO composite as a high-rate anode nanomaterial for SIBs. (1) The composite was prepared via a distinctive lattice confinement of the ternary CoMoSn-LDH precursor, and endowed with the advantageous characteristics of boosting electrochemical performances: (i) well-distributed multi-component CoS/MoS_2_/SnS heterostructures with the interface interaction between Mo and Sn cations; (ii) highly electron conductive rGO support (18.6 wt.%), (iii) appropriate specific surface area (30.3 m^2^ g^−1^) and mesoporous size distribution. (2) Consequently, the composite delivered satisfactory sodium-storage performances: a high-reversible capacity of 620.3 mA h g^−1^ after 100 cycles at 0.1 A g^−1^, and excellent rate capacity of 371.0 mA h g^−1^ after 1000 cycles at 5.0 A g^−1^; both of which surpass those of the three counterparts of sing- and bi-sulfides, as well as the non-LDH-precursor-derived multiple sulfides. (3) Furthermore, the substantial enhancement was experimentally elaborated by the low charge-transfer resistance and diffusion resistance demonstrated using EIS testing, and the dominant capacitive contribution. The results show that the LDH precursor-based synthesis route could promise an effective approach to design and prepare well-dispersed multiple-component transition-metal compounds as anode nanomaterials for SIBs.

## Figures and Tables

**Figure 1 molecules-28-05972-f001:**
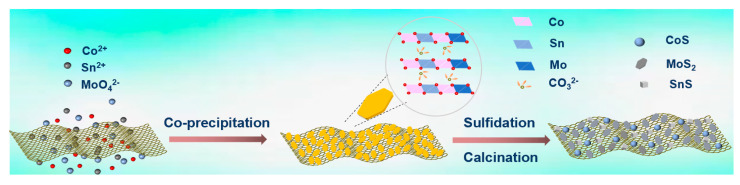
Schematic diagram of the preparation of MoS_2_/SnS/CoS@rGO composite derived from a single-source ternary CoMoSn-LDH/GO precursor. The checkmark shows the concept of cationic dispersion, which is based on the Mg/Al ordering in MgAl-LDHs [42].

**Figure 2 molecules-28-05972-f002:**
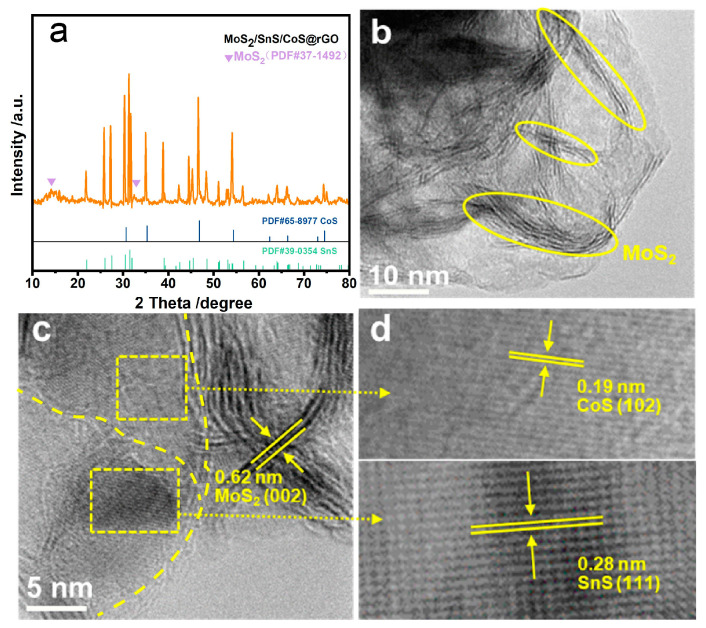
The MoS_2_/SnS/CoS@rGO composite: (**a**) XRD patterns, (**b**,**c**) TEM images, and (**d**) HRTEM images.

**Figure 3 molecules-28-05972-f003:**
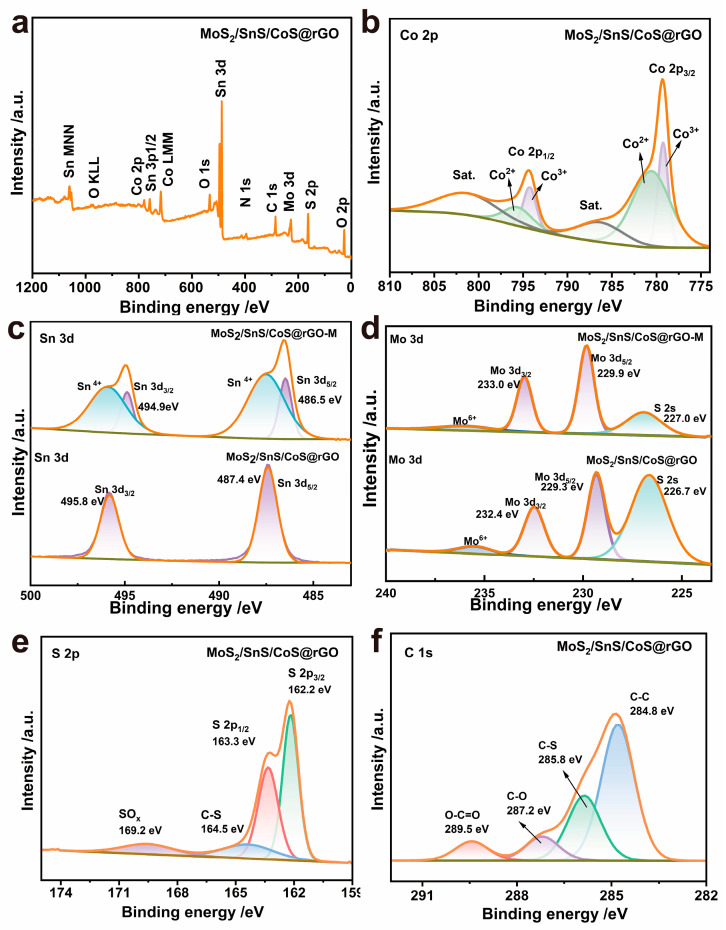
XPS spectra for the MoS_2_/SnS/CoS@rGO composite: (**a**) a full survey spectrum, (**b**) Co 2p, (**c**) Mo 3d, (**d**) Sn 3d, (**e**) S 2p, and (**f**) C 1s.

**Figure 4 molecules-28-05972-f004:**
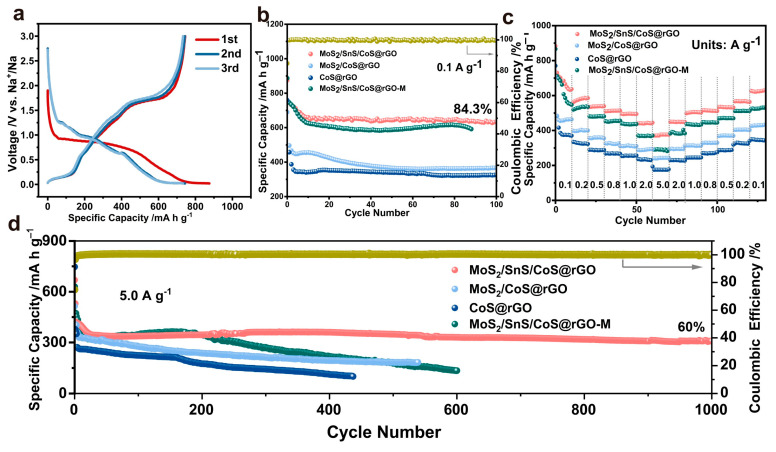
Comparison of electrochemical performance for SIBs. (**a**) The first three Galvanostatic discharge/charge profiles of MoS_2_/SnS/CoS@rGO at 0.1 A g^−1^, (**b**) cycling performance at 0.1 A g^−1^, (**c**) rate performances from 0.1 to 5.0 A g^−1^, and (**d**) long-life cycling performance at 5.0 A g^−1^.

**Figure 5 molecules-28-05972-f005:**
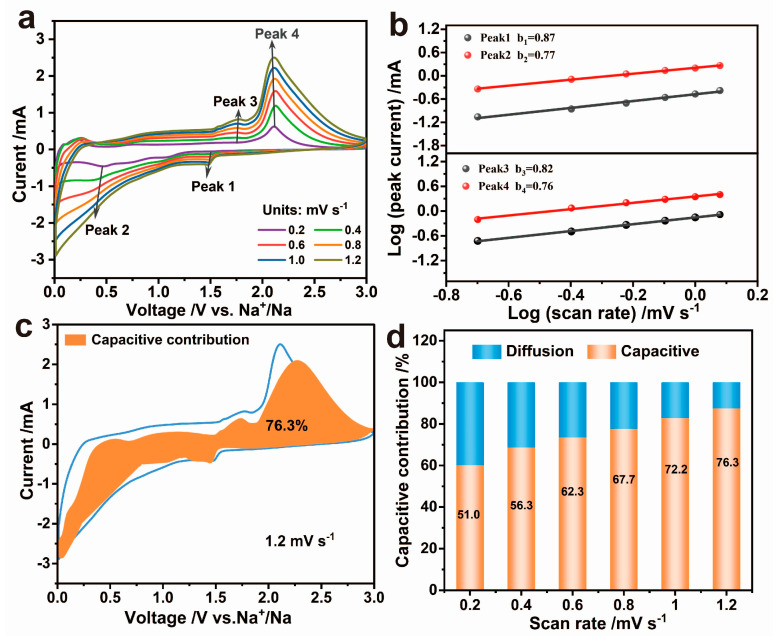
The Na^+^ kinetics analysis of MoS_2_/SnS/CoS@rGO electrode. (**a**) CV curves at different scan rates between 0.2 and 1.2 mV s^−1^, (**b**) the linear relationship between log(i) and log(ν), (**c**) capacitive contribution at 1.2 mV s^−1^, and (**d**) capacitive contributions at various scan rates from 0.2 to 1.2 mV s^−1^.

## Data Availability

Most of the data used during the preparation of the manuscript are included in the Section 2 and Section 3.

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
