# Peer review of "MoS2/SnS/CoS Heterostructures on Graphene: Lattice-Confinement Synthesis and Boosted Sodium Storage"

_molecules, 2023, doi:10.3390/molecules28165972_

Round 1

Reviewer 1 Report

There are some minor wording changes indicated in the attached mark up

The set of reactions lines (174-177) need either Gibbs or voltage values inserted

Fig S3 has an absence of Sn in a circled region which needs some explanation (Cannot seem to up load a second file but the location should be clear)

Fig S5 was stated to have scans 2 and 3 reasonably similar, but this is not the case scan number 5 or 10 should be shown for the reader to determine stability

In general the authors have done  excellent work in characterizing the materials. The improved performance of SIB is very encouraging even if long term stability still needs improvement.

Well done!

excellent only a couple of typos as highlighted in attachment

Author Response

We especially acknowledge the reviewers for the valuable comments; all of which help boost the quality of the revised manuscript significantly for publication.In accordance with the comments from the reviewers, we carried out the supplementary experiment, and added the appropriate discussion and analysis, as well the sample preparation schemes into the revise manuscript. All changes have been highlighted in yellow in the revised manuscript.

Attached please kindly find our point-by-point responses to the reviewers’ comments.

Reviewer 2 Report

Comments and Suggestions for Authors

Dear Authors,

The Title:

MoS2/SnS/CoS heterostructures on graphene: lattice-

confinement synthesis and boosted sodium storage

I have to read your manuscript with great attention and interest.

The authors of the manuscript worked on the development of high-performance, multi-component composite anode nanomaterials for sodium-ion batteries (SIBs). The heterostructural composite dispersed on graphene was prepared as anode nanomaterial for SIB. The characteristic net-closing effect of the ternary double hydroxide CoMoSn (CoMoSn-17 LDH) as a precursor was observed.

The submission falls within the scope of the journal and is sufficiently original. I recommended the publication after MAJOR REVISIONS.

General comments:

  • What is the strengthening observed for the tested materials? I suggest adding a diagram explaining the effect.

Specific comments:

  • Doesn't Fig. 1 need reference number 42?
  • There is a checkmark in Fig. 3, what is it for?
  • On what basis was a scan rate of 0.1mV s-1 considered the best for research? Measurements were made at other potential sweep rates - Fig. S5
  • Add response numbers, line 174-177
  • What information was obtained by analyzing the pore size distribution? What was it used for?
  • I suggest to chapter 3.1. and 3.2. add a sample preparation scheme
  • Conclusion make in the form of points.

Author Response

(The authors gave the same response as above.)

Round 2

Reviewer 2 Report

I would like to thank the authors of the manuscript for their explanations and for responding to my comments. I recommend the revised manuscript for publication.